# Current Status of Omics in Biological Quality Elements for Freshwater Biomonitoring

**DOI:** 10.3390/biology12070923

**Published:** 2023-06-28

**Authors:** Jorge Machuca-Sepúlveda, Javiera Miranda, Nicolás Lefin, Alejandro Pedroso, Jorge F. Beltrán, Jorge G. Farias

**Affiliations:** 1Doctoral Program on Natural Resources Sciences, Universidad de La Frontera, Avenida Francisco Salazar, 01145, P.O. Box 54-D, Temuco 4780000, Chile; 2Department of Chemical Engineering, Faculty of Engineering and Science, Universidad de La Frontera, Temuco 4811230, Chilealejandropedroso7@gmail.com (A.P.);

**Keywords:** freshwater biomonitoring, omics approaches, biological quality elements, biomarkers, bioindication, ecotoxicology

## Abstract

**Simple Summary:**

Freshwater ecosystems face various threats, especially in recent decades, that pose unprecedented challenges to human health, water supply, agriculture, forestry, ecology, and biodiversity. Although progress has been made in biomonitoring techniques tailored to specific countries and communities, significant constraints exist in assessing and quantifying biodiversity and its interaction with detrimental factors. The incorporation of modern techniques into biomonitoring is challenging, with multiple perspectives. This review aims to provide a comprehensive overview of contemporary advancements in freshwater biomonitoring, focusing on omics methodologies such as genomics, metagenomics, transcriptomics, proteomics, metabolomics, and multi-omics. Additionally, this work highlights the need for modernization by presenting case studies, examining studied organisms, and evaluating the benefits and drawbacks of these methodologies. The utilization of advanced high-throughput bioinformatics techniques represents a departure from conventional practices, necessitating a significant shift. The contributions of omics techniques to biological quality elements (BQEs) and their interpretations of ecological problems are crucial for biomonitoring programs. These contributions identify interactions between different levels of biological organization and their responses to specific critical conditions.

**Abstract:**

Freshwater ecosystems have been experiencing various forms of threats, mainly since the last century. The severity of this adverse scenario presents unprecedented challenges to human health, water supply, agriculture, forestry, ecological systems, and biodiversity, among other areas. Despite the progress made in various biomonitoring techniques tailored to specific countries and biotic communities, significant constraints exist, particularly in assessing and quantifying biodiversity and its interplay with detrimental factors. Incorporating modern techniques into biomonitoring methodologies presents a challenging topic with multiple perspectives and assertions. This review aims to present a comprehensive overview of the contemporary advancements in freshwater biomonitoring, specifically by utilizing omics methodologies such as genomics, metagenomics, transcriptomics, proteomics, metabolomics, and multi-omics. The present study aims to elucidate the rationale behind the imperative need for modernization in this field. This will be achieved by presenting case studies, examining the diverse range of organisms that have been studied, and evaluating the potential benefits and drawbacks associated with the utilization of these methodologies. The utilization of advanced high-throughput bioinformatics techniques represents a sophisticated approach that necessitates a significant departure from the conventional practices of contemporary freshwater biomonitoring. The significant contributions of omics techniques in the context of biological quality elements (BQEs) and their interpretations in ecological problems are crucial for biomonitoring programs. Such contributions are primarily attributed to the previously overlooked identification of interactions between different levels of biological organization and their responses, isolated and combined, to specific critical conditions.

## 1. Introduction

Keck et al. [1] are noticeably clear concerning the modern conceptualization of freshwater biomonitoring. Modern conceptualization involves the use of bioindicators designed as biological quality elements (BQEs), such as fishes, macrophytes, macroinvertebrates, benthic diatoms, and phytoplankton, along with hydromorphological and physicochemical measurements. Freshwater biomonitoring programs combine practices to understand ecological conditions in ecosystems, particularly in BQEs, through biological communities, especially under anthropogenic disturbances [1]. The crucial characteristics of biomonitoring are rapidness and robustness, which produce reliable information for management and mitigation actions in the context of environmental degradation [2]. How freshwater ecosystems are influenced, referring to the distribution of organisms due to human-induced changes to habitats, is paramount. We must allow for their protection to preserve their benefits for humans and the environment and advance in conservation biology [3]. The most worrisome factor of environmental degradation is the decline in species diversity at unprecedented rates, which has been addressed precisely through biomonitoring programs, a tool with ecological, simple, and economical attributes [4,5]. Biomonitoring programs are decisive in assessing impacts based on BQEs’ key concept, which estimates their variability on ecosystem structure [6].

Given the high fragility of freshwater ecosystems in view of human disturbances and stressors, biomonitoring takes on the importance of threats that have had repercussions for centuries (e.g., organic pollution), others more recently (e.g., acidification), and likely more in the future (e.g., nanoparticles) [7]. The most prominent threats that cause devastating effects are climate change and global warming, with their consequent temperature and physicochemical anomalies [8]. Water pollution is one of the most widely researched stressors, with estimates and assessments of the general status of aquatic environments being conducted through ecological and chemical indicators [9]. Pollution by pesticides or chemical activities for agriculture [10], pharmaceuticals [11], heavy metals [12], wastewaters [13], and microplastics [14] are the most troublesome impacts caused by this factor. Other pollutants, specifically personal care products, endocrine disruptors, and nutrient enrichment, have recently gained importance in biomonitoring [15,16]. The alterations caused by dams, land use changes, invasive species, and urbanization are also relevant in the biomonitoring horizon [17,18,19,20].

BQEs in freshwater biomonitoring mainly comprise two types of biological communities: diatoms and aquatic insects, as well as others such as fish, zooplankton, and phytoplankton. Diatoms are microscopic unicellular algae presented mostly in benthic habitats (phytobenthos) and are highly diverse, with more than 12,000 species described [9]. It has been characterized by specific ecological preferences regarding pollution gradients [21], physicochemical alterations [22], and toxic industrial wastes [23]. The diversity of diatom assemblages can be estimated in certain ecosystem processes, such as resource use efficiency, which can be affected by flow intermittency and varying degrees of pollution [24]. The phytoplankton community is frequently monitored to detect environmental impacts. Similar to the periphyton, they have similar environmental requirements and dispersal capacity [25,26]. Aquatic invertebrates are widely used as indicators of ecosystem health because of their limited mobility, life cycles in water, and response to anthropogenic pressures [27]. Within the group of macroinvertebrates, these organisms are the most common in the classification of ecological integrity based on biotic indices [28]. Aquatic insects, such as Ephemeroptera, Plecoptera, Trichoptera (abbreviated as EPT), and Diptera, compose the core of the ecological food web. They feed on producers, are preyed upon by higher consumers, and have been the leading groups for biomonitoring studies and river management [29]. In summary, diatoms and aquatic insects are among the most important groups in running water biodiversity, but relatively few studies have investigated their complex relationships [30]. Concerning fish and other organisms such as zooplankton, heterotrophic communities are recognized as good indicators of ecological stream conditions due to their ubiquitous presence, high responsiveness to physical and chemical parameters, and obvious impact on other communities within food webs because of their carnivorous and herbivorous activities [31,32].

The myriad of existing threats to the communities described leaves a scenario with vast unexplored areas in biomonitoring knowledge, which involve a series of challenges. Biotic indices have several limitations related to bioindicators and biomarkers, such as not utilizing the combined effects of multiple disturbances and stressors in freshwater ecosystems [19]. The taxonomic hurdle is a general agreement in several studies, mainly due to morphological misidentification analysis, time-consuming high-quality microscope requirements, the need for skilled taxonomists, and impediments in reaching species-level identification. Another issue is related to sampling procedures by nets, which leads to incomplete community sampling. It takes prominence in the ability and skills of investigators and the accessibility of sampling sites, with consequent damage to natural habitats and organisms [27,28,29]. Furthermore, in a deep analysis, biomonitoring cannot decipher the stenotopic and eurytopic attributes of organisms in the presence of a given number of stressors on a local scale. The above is related to cellular mechanisms and metabolic, behavioral, and physiological settings, which have been studied by biomonitoring [33,34] but have not directly unveiled the effects of environmental stressors at these levels.

The omics techniques and molecular tools for freshwater biomonitoring appear to gain a fuller understanding of the ongoing processes on all levels of biological organization. It allows the development of a methodology for enhancing environmental status in the presence of particular stressors [35]. Omics approaches are applied in ecotoxicology to find biomarkers to elucidate toxicity mechanisms. The best-developed methods include transcriptomics, proteomics, and metabolomics [10,36], but their applications are still comparatively expensive compared to traditional methods. The genomics tools represent a promising alternative to taxonomy-based methods and exceed the limitations of morphology-based species determination [37]. Additionally, it has been registered that metabarcoding and metagenomic studies have led to biomonitoring purposes because of the enormous increase in available genetic data for organisms, communities, and habitats. This is relevant for comparing the performance of morphological and metabarcoding methods [38,39]. Therefore, it is evident that genomics, proteomics, and metabolomics are new tools for finding biomarkers that reflect the effects on biological communities by chemical exposure. For example, these omics disciplines apply high-throughput methodologies to simultaneously assess changes in the expression of hundreds to thousands of genes, proteins, or metabolites [40].

Freshwater diversity assessment through biomonitoring is a key element for ecosystem management in an alarming context, a product of climate change and human stressors. An enabling element has been the BQEs, which allow for the more rapid acquisition of diverse data and knowledge of molecular processes that underlie development in different impacts and ecological indicators status scenarios. This review aims to explore the topics of omics applications specifically to modernize freshwater biomonitoring, involving genomics, metagenomics, transcriptomics, proteomics, metabolomics, and multi-omics. Study cases of the omics approach in freshwater ecosystems in diatoms, aquatic insect communities, phytoplankton, macrophytes, fishes, and zooplankton will be reviewed. We highlight the current status of biomonitoring through a state-of-the-art synoptic review of the different types of omics that have been theorized and performed. The second part is the description of perspectives and challenges on the use of omics in freshwater biomonitoring schemes. The advantages and disadvantages of feasible opportunities reported in this matter concerning the need for a multifaceted framework in scenarios with more aggressive impacts and data management are tackled.

## 2. State-of-Art of Omics in Freshwater Biomonitoring

### 2.1. How Biological Quality Elements (BQEs) Can Be Used in Omics

The sights related to BQEs used in freshwater biomonitoring practices, and their connotations in the contexts awarded by omics approaches, only for fish, macroinvertebrates, macrophytes, microalgae, and zooplankton communities, are going to be explained. To begin with, based-omics biomonitoring can be classified into lower (cells, tissues, etc.) and higher (e.g., communities) levels of biological organization analysis [41]. As an interface, a third expected level of analysis would be at the behavioral [42,43] level. Prone to dimensional features, macroinvertebrates, and fish communities are typically utilized in analyses at higher levels of biological organization, with an emphasis on field-based assessments. In contrast, microalgae and zooplankton communities are considered in lower-level analyses, which usually consist of controlled experiments such as bioassays. Therefore, toxicological procedures have been extended to these communities, which have been studied in several experimental efforts based on cultures [44,45,46,47,48,49] in pursuit of novel biomarkers or more robust omics-response information about them. Concerning communities, bioindicators have been less studied by omics approaches than biomarkers, although employing model species for specific processes is common. These species are typically laboratory-bred, well-studied, and characterized [50]. For instance, in genomics, *Daphnia magna* is usually used as a model organism to study biomarkers based on genes involved in biochemical defense against toxicants [51]. The significant issue about community-level exploration is whether it can be utilized to its full potential by omics lens, specifically proteomics [52]. This is because ecological processes are difficult to tackle due to the increasing acquisition of big data and more complex organizations.

Modern tools provided by molecular techniques are precise in improving sampling and identification, emphasizing smaller and microscopic organisms such as microalgae. The traditional or morphological analysis is more profound than is generally thought because the life cycle, particularities, and impacts of environmental factors affect morphological aspects in some organisms. However, this is still neglected in biomonitoring studies and water quality assessments [53]. On the other hand, innovative high-throughput sequencing (HTS) techniques abound with a high degree of modernization, solving the inefficiency of identification and placing more attention on providing more information at greater resolution, depth, consistency, and at a lower cost than morphological methods [54]. HTS technologies are sequencing techniques that allow for the simultaneous analysis of millions of sequences, compared to the Sanger sequencing method, which processes one sequence at a time, and they are qualified to sequence multiple DNA molecules simultaneously [55,56]. The bottleneck of taxonomic identification has repercussions on biodiversity measurements and the large number of sampled data over a long-term period. This issue is compounded by unknown introduced species, which can lead to misleading conclusions about their response to stressors [57]. For this, improved biomonitoring for a big data scenario is fundamental through innovative HTS techniques, ensuring the decision-making power for detecting and controlling the stressors more rapidly and simultaneously over a large spatial area [6].

At lower levels of biological organization, omics approaches are focused on the search for new biomarkers in ecotoxicological frameworks for a few species/taxa, targets of metabolic pathways in cells/tissues, or patterns of microbial communities in response to environmental impacts along the source-to-outcome pathway (STO) [58,59]. Single-cell omics (in conjunction with meta-omics techniques, described below) is a method that covers scales from the individual microbe to the community level in the same environmental sample, focusing on microorganisms. This allows for connecting the resulting information from each ecological scale [60]. In microalgae, physiological changes due to pollution (such as Cu) can be assessed by observing morphological alterations such as irregular motility, physical deformations, or growth rate alterations [61]. At once, whole-genome sequencing, transcriptomics, proteomics, and metabolomics have revealed a detailed mechanism of lipid catabolism and anabolism, which are related to various structural and growth types [62]. For example, outside the diatom case, also in aquatic insects [63], teleost fish [64], and other vertebrates [65] in lipid metabolism have been developed. Similarly to these examples, diatom molecular biology has been incompletely epitomized with prominence in changing environmental physiological responses, despite the development of genome sequencing, RNA-seq, and metabolic pathway data acquisition in recent years [66]. Biofilm studies can be enhanced through transcriptomics or proteomics analysis to identify biofilm physiology and to detect the source attribution of pathogens via tracing the contamination route, by the response of the embedded cells, or from matrix material [67]. According to [68], involving plankton dynamics, which are at a higher level than unicellular organisms, meta-omics techniques have great potential to provide deep knowledge of changes in biological and functional diversity and, therefore, plankton trophic network data. To eke out with respect to ecological scale results, single-cell omics methods, together with meta-omics approaches, could be powerful in aiding environmental policies. They provide an overall understanding of the temporal dimension of affected or unaffected regimes in all aquatic habitats.

Omics approaches must be more appropriate for physiological studies than behavioral studies, although techniques within the omics framework related to behavior-based aims are challenging. A review explains that omics approaches, such as genomics and transcriptomics, can be useful for unraveling the behavioral aspects of fishes, such as migratory behavior [69]. Behavioral variables can serve as intermediates between chemical analysis and monitoring tools in streams, as demonstrated in the case of the sexual behavior of fathead minnows (*Pimephales promelas*). It was found that specimens exposed to sewage treatment plants could induce a site-specific gene expression pattern in their gonads and livers [42]. Nearly the work related to behavioral aspects would be extended to fitness-based and cellular-level traits, for example, the case of mode of action (MOA) dynamics. MOA can be defined as a functional change at the cellular level, which has already been evaluated in aquatic organisms such as cladocerans threatened by cyanobacteria toxicity through transcriptomics analysis [70,71]. In fact, an MOA-aquatic toxicity endpoint database (MOAtox) that meets USEPA standards already exists. It consists of four species (*Daphnia magna*, *Lepomis macrochirus*, *Oncorhynchus mykiss*, and *Pimephales promelas*), representing 22% of the 300,000 ECOTOX records [72,73]. Concerning macroinvertebrates, in the current context of multiple environmental stresses that deplete oxygen quantities in water [74], it is important to clarify how behavioral adaptations at the genetic or enzymatic levels are paramount to biomonitoring practices based on key functional traits. The case of dissolved oxygen involves both respiratory and molecular responses. The latter is a remarkable example of hypoxia, where genes and enzyme expressions are conducted, which affects behavioral adaptations in BQEs [75]. The ethological analysis would be interesting when covered with omics at an individual level, such as neurophysiological studies on rare behaviors. For example, death-feigning, immobility behavior, inactivation processes (diapause), camouflage, and autotomy, among others, are frequent in some macroinvertebrates and fish. In the following sections, we review the existing omics technologies in the context of freshwater biomonitoring through several study cases. The advantages and disadvantages of each are summarized in Table 1.

### 2.2. Genomics

The domain of omics techniques involves the physiological and genomic knowledge of species. It is precisely used to analyze the altered metabolic pathways by stressors via the simultaneous quantification of a set of genes involved, expressed genes (transcriptomics), proteins (proteomics), and metabolites (metabolomics) [107]. First, genomics is constituted by tools that improve the resolution of taxonomic information and refine the detection of cryptic, rare, or new non-native or invasive species [108]. The environmental genomics tools agree with replacing morphology-based methods for taxonomic determination with enhanced options [37]. For example, the taxonomic distinction between pennate and centric lineages of diatoms has been clarified through comparative genome sequences, revealing molecular diversification on a genome-wide level [109]. Via genomics, it is conceivable to investigate seasonal carbon dynamics through diatom growth and photosynthesis rates. This is because certain genes (found in the nuclear genome) encode proteins involved in the mechanisms of carbon uptake maximization [76]. Furthermore, genomic data has been utilized to explore the local adaptation of non-model macroinvertebrate species in stressed environments, supported by the rise of high-throughput sequencing. Whole-genome sequencing has become an increasingly cost-effective research tool [77,78]. In macroinvertebrates, the genome sizes of holometabolous and hemimetabolous orders have been studied, revealing a small genome for the former and a large genome for the latter. The Odonata order falls between these two groups in the intermediate size range [110]. Comparative genomics has detected the functional similarities between non-model and model (most sensitive or bioindicator) species within ecotoxicological biomonitoring routines to identify toxicity pathways and novel molecular biomarkers in species that have not been defined yet [79]. However, there is a dramatic lack of data on freshwater organisms in digital repositories, such as the genomes of aquatic insects, limiting research progress in the field at both fundamental and applied scales [77]. 

Regarding fish, they are the most attractive group investigated for gene expression at a highly advanced level, and with a high number of chemical/pharmaceutical stress elements such as metals, metalloids, nanoparticles, single organic compounds, endocrine disrupting compounds, mixture toxicity, effluent toxicity, and sediment toxicity, which affect metabolism, fertilization, cellular processes, amino acid synthesis, apoptosis, among others [111]. Certainly, genomics deepens insights into eco-evolutionary dynamics at the population or community levels because the gene expressions that are significantly up- or down-regulated show a link to the evolutionary process of the defense of a given trait [112]. Complementing Table 1, here are some additional examples of species with complete genomic records: *Margaritifera margaritifera* [113], *Paedocypris progenetica* [114], *Acipenser ruthenus* [115], and *Orestias ascotanensis* [116].

### 2.3. Metagenomics

Metagenomics analysis emerged from microbial studies as an advanced molecular method for exploring the taxonomic and functional diversity of organisms sampled from the natural environment. It is also used to determine the genome sequences of uncultivable and rare microbes, describe microbial ecosystems, and discover novel genes and gene products [117]. In the case of invertebrate communities, the Ecobiomics Project from Canada’s Genomics Research and Development Initiative (GRDI) has used the term “zoobiome” to refer to all the individual invertebrates, their genes, and habitat occurring in a particular place (the benthos of a particular stream area). This concept is useful for advancing and expanding environmental assessment developed by metagenomics [118]. For classic examples, metagenomics and specifically the mitogenome (mitochondrial genome) have been proven to estimate the individual biomass of macroinvertebrates, achieving more informative predictions of biomass content from bulk macroinvertebrate communities than metabarcoding [80]. Although mitochondrial genomes may also be suitable for bio-assessment, it is a priority to have further validated and completed mitochondrial reference genome libraries [38]. With metagenomes, it is possible to estimate approximately the biogeochemical processes in estuaries. This has been investigated through the metagenomes of diatom detritus from sediments as allochthonous organic matter that induces changes in microbial communities. These changes, in turn, influence organic matter degradation and recycling [81]. Genome skimming is an efficient approach that consists of superficial shotgun sequencing to obtain genomic data from eukaryotic organisms from high-copy portions of nuclear DNA and organellar DNA of a single specimen. This may help metagenomics studies due to improved genomic databases [119]. It has been reported that genome skimming has been used in meiofauna, specifically for copepods, because they are tiny in size and belong to cryptic taxa with intraspecific polymorphism, which poses many hurdles. 

Additionally, decapod crustaceans have been studied through nuclear and mitochondrial genes to support more robust and comprehensive phylogenetic analyses [119,120]. Metagenomic biodiversity assessments for invertebrates can also be developed, but there is a lack of protocols that reduce the time of sample processing without the risk of cross-contamination [121]. The pros of metagenomics are mainly to give an overview of the metabolic potential and reconstruct whole-genome sequencing (WGS) to include the taxa with small genomes in simple communities. The cons are that pathway genes or markers of interest may only be recovered at low frequency, and taxonomic assignments based on markers are not reliable [82].

### 2.4. Transcriptomics

At the beginning of the past decade, next-generation sequencing technologies (NGS) were utilized for almost every DNA-based molecular field. Transcriptome analysis was not an exception, with functional transcriptomics progressing through both microarray technology and RNA-Seq, measuring absolute transcript levels of both sequenced and unsequenced organisms [122]. With the improvement of transcriptomics techniques, their use in ecotoxicology has been accomplished, as acquired knowledge about changes in environmental conditions and exposure to pollutants can influence the expression of gene transcripts [83]. As an example of ecotoxicological freshwater bioassessments at the transcriptomic level, studies have used *Dreissena polymorpha* as a model organism to evaluate a broad range of hazardous chemicals in laboratory and field studies. Specific insecticides, such as imidacloprid, have also been studied for their effects on certain species of mayfly nymphs, causing stress [83,84]. 

Moreover, at the transcriptomics level, it is possible to analyze comparative transcriptome-wide searches for genes of catfish species related to adaptation to high-elevation environments and, at the same time, identify their associated functions [85]. Other specific adaptations regarding the life cycle of aquatic insects have been studied. For example, through the Coleoptera larval and adult transcriptomes from *Aquatica leii* (freshwater) and *Lychnuris praetexta* (terrestrial), it has been assessed whether certain genes have undergone adaptive evolution to freshwater, associated with metabolic efficiency and morphological processes [86]. Likely, transcriptomics can be associated with certain protein expression analyses originating from molecular responses during hypoxia events (hypoxia-inducible factor complex). This has already been studied in stonefly species. Hypoxia affects the transcription of genes or gene regions associated with oxygen-related proteins [77]. Through transcriptome exploration, it has been possible to shed light on new biomarkers by expressed sequence tags (ESTs) or via pilot bioassays for heavy metal environmental remediation and impact assessments, which have been conducted with the mollusk *Physa acuta* and the green alga *Chlamydomonas reinhardtii* [44,87]. Dietary aspects of invertebrates and meiofauna exposed to toxic algae have been reported through transcriptomics analysis to recognize the effect of genes and presumably biological pathways [71,88]. In addition, transcriptomics has also contributed to unveiling the metabolic alterations in algae species in bioassay contexts to certain macrolide antibiotics such as clarithromycin and roxithromycin [48,100].

### 2.5. Proteomics

For protein biomarkers, proteomics provides a general recognition of structure (in the sense of changes at the molecular level close to the organismal phenotype), functional cellular information, response mechanism, and multiple changes in molecular processes simultaneously in the presence of organic or inorganic pollutants [5,33]. The proteins can be considered as molecular signatures of species, which may be useful in species monitoring within heterogeneous ecosystem biomass (relevant in a trophic context) and serve as indicators of ecosystem state [123]. Furthermore, biomonitoring surveys must be improved by understanding the physiological changes associated with behavioral monitoring systems. Proteomic analysis has been utilized for clam-based model species [89]. For example, the grazing behavior of aquatic insects is a mechanism that affects algae communities (bottom-up processes) [30]. Hence, proteomic analysis would be appropriate to estimate not only the physiological aspects of different taxa but also ecological processes. This, as well, is associated with variations in the food chain due to alterations in amino acid abundances and proteomes. These alterations have been detected in algae (specifically, the genus *Chlorella*) after exposure to metals [90]. 

Regarding diatoms, to date, the ecotoxicological perspective is predominant concerning the studies focused on the stress effects and responses by proteomic changes, but only for individual stress, not multiple stresses [91]. Certain aquatic insects (Chironomidae family) have also been used to determine the impact of natural insecticides, such as Spinosad, on their metabolism, specifically the alterations in globin protein abundance [92]. Proteomics might reveal tolerance mechanisms of metal exposure on macrophytes related to phytoremediation through differential gene expression or post-translational modifications of proteins. This can clarify the level change in proteomes with an efficient response [124]. Recognizing the biological functions of proteins or a group of proteins (functional proteomics) is also adequate for studying the acclimation response of algae organisms to different environmental conditions. Algae are related to light dynamics, nutrient conditions, and families with large genes that encode pigment-binding polypeptides [125]. The main advantage of proteomics would lie in being more appropriate for recognizing the effect of plastic responses to the variability of environmental conditions and selective pressures, such as rapid human-induced climate change [33]. Temperature alterations may severely affect organisms’ homeostasis and physiological status, such as fish. At the same time, it can also affect the toxicity of pollutants on aquatic organisms [126,127]. Proteomics analysis is precise for understanding the general stress caused by drastic temperature alterations and for long-term exposure to polystyrene microplastics, which are frequently detected in environmental samples. These microplastics have been noticed to affect the fitness of daphnids by decreasing and increasing multiple groups of their proteins [93].

### 2.6. Metabolomics

Metabolomics has been used for evaluating the effectiveness of freshwater bioindicators, but not much in the use of biomonitoring practices [41]. Metabolomics is now a well-established scientific field in systems biology. It estimates changes in metabolic biochemistry (through untargeted metabolomics) by low molecular weight metabolites (molecular weight < 1000 Da). These metabolites are, in part, the product of gene expression and protein translation and represent the most functional measure of an organism’s physiology and response to toxic stress [128,129]. In pursuing biomarkers, metabolomics is governed by two main approaches: targeted and non-targeted. Targeted metabolomics involves quantifying one, or a set, of known metabolites, which are usually related to a specific pathway or biological activity. The non-targeted approach is a fast, high-performance analysis based on the data of all metabolites present in a given set of samples without prior knowledge [130]. Analytical advances allow for the determination of metabolites in small organisms, such as *Daphnia* spp., in the face of sublethal contaminant exposure [95]; this is a great accomplishment for freshwater biomonitoring, where frequent capture and study of small organisms is necessary. In ecotoxicology, metabolomics analyses are feasible for evaluating the toxic effect of nanoparticles. For example, in *Scenedesmus obliquus*, such analyses have yielded advantageous results in stress response for low concentrations and long-term exposure to certain fullerenes [46]. The toxicological effects of persistent micropollutants, such as carbamazepine and methylmercury, have also been studied independently and together through metabolomics analysis applied to the bivalve *Dreissena polymorpha* [96]. Interactive effects of stressors (Cu and warming) have also been investigated in the microalga *Scenedesmus quadricauda* (Chlorophyceae), using metabolomics techniques and physiology simultaneously [97]. Following the algae species, the diatom *Stephanodiscus hantzschii* was evaluated under enriched nutrients but at different temperature conditions (5 °C and 15 °C), where colder temperatures suppressed population growth through the deactivation of various internal metabolisms [98]. Moreover, the chemical pollution effect from emerging contaminants (such as pharmaceuticals and endocrine-disrupting compounds) on aquatic invertebrates has been assessed through non-targeted metabolomics in natural conditions (in situ experiments), specifically in caddisfly larvae [99]. Similar to proteomics, metabolomics has been utilized to evaluate distinct types of tissues of the crayfish *Faxonius virilis* to ascertain the efficacy of this approach as a short- or long-term biomonitoring tool [131]. For fish, metabolomics methods can reveal the mechanisms related to metabolic alterations produced by multiple environmental pollutants, which were realized through non-lethal methods, specifically in muscle tissue [132].

### 2.7. Multi-Omics

The multi-omics techniques used to estimate the health of freshwater ecosystems, such as metagenomics and metabolomics, can reveal bacterial diversity responses in an urban stream system [133]. Metabolomics and transcriptomics have been useful in investigating reactions at these levels in algae species in the presence of different concentrations of the antibiotic macrolide clarithromycin [100]. An innovative study explored the resistome (antimicrobial resistance levels of microbial communities) found in European lakes using multi-omics techniques, associated genomics, and metagenomics for comprehensive analyses of important groups of antibiotics [101]. According to [102], metabolomic, proteomic, and transcriptomic data can be integrated with adverse outcome pathways for ecotoxicological procedures to link sub-individual biomarker responses and potential effects at higher levels of biological organization. Mechanisms of action through which contaminants achieve adverse outcomes at higher biological organization levels can be useful, as provided by multi-omics. This has been addressed complementarily with environmental metabolomics in the context of certain contaminants, such as pharmaceutically active compounds in water [103]. Ecosurveillance methods have been developed as part of biomonitoring programs to examine microbial diversity and metabolism in estuary sediments through vegetation gradients, with metabolomics, transcriptomics, and metagenomics as central pieces [134]. In addition, the negative impact of methylmercury and inorganic mercury in aquatic food webs has been estimated through the bioaccumulation on the macrophyte *Elodea nuttallii* using transcriptomics, proteomics, and metabolomics analysis. This revealed a direct effect on a higher number of genes and in the cytosol [104]. Regarding proteomics, ecotoxicoproteomics needs multi-omics approaches due to developing further molecular information and constructing the most suitable databases for protein identification and pathway analysis in non-model species [94]. For more detailed discussions on different proteomics integrations with other omics techniques in freshwater ecotoxicology, see [52].

## 3. Use of Omics on Freshwater Biomonitoring: Perspectives and Challenges

### 3.1. Omics in Freshwater Biomonitoring: A Multifaceted Framework

Here, we discuss the perspectives and challenges of using omics techniques in various BQE scenarios, considering both recently discovered and well-established environmental pressures. First, it is important to consider that a basic framework in biomonitoring based on molecular techniques consists of stages such as sampling/collection, preservation, and extraction of biological material. Regarding the sampling stage, modern biomonitoring based on non-destructive DNA barcoding and metabarcoding performances contributes to identifying additional taxa and provides posterior morphological analysis [135]. Additionally called “high-resolution” diversity assessments, DNA-based approaches provide detailed evidence for understanding the changes in community composition and their potential causes and consequences better than tissue-based approaches [105]. The principle of non-destructive approaches is to transfer DNA molecules from organisms to ethanol for preservation without the need for tissue crushing to obtain bulk samples. 

Additionally, it is possible to obtain stomach contents through regurgitation, providing a more detailed insight into the community [3]. Obtaining more data in a time-efficient manner while circumventing the destruction of organisms is similar to what was accomplished by Rivera et al. (2021) [136]. They obtained acceptable results in detecting macroinvertebrate DNA from biofilms on the upper surface of stones, thereby strengthening the findings on the diversity of cryptic species that are hard to acquire through traditional methods. In addition, the use of environmental DNA (eDNA) as a biomonitoring method that can be used for the sampling of diatoms, macroinvertebrates, and multiple taxonomic groups from aquatic ecosystems in a non-destructive framework [137] arises mainly due to the limitations and bottlenecks described above (sampling bias, taxonomy skills, and time inefficiency) [30]. Furthermore, the eDNA method has been used to propose many objectives for diversity measurements in multiple taxa [138,139,140], integration of ecological status assessments [141], invasive alien species monitoring [108], and hypotheses such as the rivers being a kind of “biodiversity conveyor belts” across terrestrial and aquatic ecosystems [142].

Contrasting the classic and modern methods described above, it is clear that there is a trend toward sequencing and omics techniques. The latter confers, in the case of metagenomics, a new view of the wide species diversity of macro and microorganisms in diverse environments [143]. This is necessary for freshwater biomonitoring practices, such as via microalgae. Omics are a cornerstone analysis of the effects of human stressors on freshwater ecosystems, mainly at the molecular level. This corresponds to an indirect inference for the most important objectives of biomonitoring studies [79]. Related to metagenomics and genomics, affordable methods are utilized for biomonitoring. However, a multidisciplinary understanding that represents all biological scales, from genetics to ecology, is needed. It has been unraveled which genomic methods are appropriate for exploring the adaptability and resistance of aquatic populations to impacted environments [77]. An important question mark in which four responses establish how to realize certain management, conservation, or restoration modus operandi to somehow overcome these impacts. The interest in omics techniques and their combinations with each other appeared in 2008 to obtain a broader picture of biological systems (from organism to community) and expand the knowledge provided by a single study [144]. The proteomics, transcriptomics, and metabolomics approaches can provide information about biomarkers (including further exploration of known biomarkers or discovery of new ones). Knowledge of adaptation mechanisms in molecular or biochemical response dynamics is necessary to acquire [5,41,86]. In addition to the biodiversity data, omics data can likely fulfill and operate with the same characteristics. This considers the resulting data from sampling, the availability of archived data and workflows for analysis, and the constraints and challenges in each [145]. Eventually, the omics techniques are recognized as technologies that have emerged from laboratory performances, which can be partially transformed for application in environmental conditions. Namely, it is favorable since modern research into bioindication and biomonitoring should aim to ensure the comparability of effects determined in the laboratory and the field [146].

Within the framework of ecotoxicology and stress ecology, subdisciplines of biomonitoring that can be joined to omics protocols, the increasingly complex chemical structure of toxic compounds or pollutants that arrive and accumulate in any aquatic environment cause more entangling changes at the gene or protein expression level. Thus, it is concluded that conventional toxicity tests are insufficient to evaluate the total environmental risk [147]. The degree of functionality of omics data is also an interesting attribute that can signify an inflection point. It is extended to a wide variety of species based on standardized functional annotations [148]. It cooperates with the clarification between tolerance and resistance terms regarding genetic or physiological adaptation in the presence of contaminants. Apart from the relevance of biological knowledge per se, integrating these data, which emerged from omics techniques, is also crucial due to creating a general view related to the dynamics that occur in the whole biosystem rather than in their individual compartments [149]. 

For this reason, the modernization of ecotoxicology must ensure sophisticated alternatives to the mortality endpoint in bioassay measurements to assess environmental quality, and omics approaches provide this. For example, in terms of sensitivity, the measurement of metabolomes over the course of a determined bioassay in a population or community can be an estimate of chronic chemical pollution, generating changes in the metabolomes of aquatic insects [104]. This will clearly depend on the species with which the analysis is performed. In the case of the isopod *Asellus aquaticus*, no significant differences in Na+/K+-ATPase were noted in the presence of high levels of trace metals. This presumably indicates an increase in total glutathione levels, indicative of reactive oxygen species production. Therefore, the organisms have adapted to the polluted environment by producing antioxidants to combat the increased oxidative stress. This point should be carried out in more studies to determine if this could lead to a reduction in bioassay effectiveness over time, which also depends on the species and type of pollutant [150].

The multifaceted framework for omics technologies in freshwater biomonitoring is an essential attribute to proposing systematized protocols for sampling/collection, preservation, and extraction stages. As molecular approaches advance, certain omics technologies, such as metagenomics, can help complement modern studies focusing on microbiota and microbiomes. These technologies are useful for comprehensively analyzing microbial communities and complement traditional studies [117]. Given that they have emerged from surveys focused on microbial communities, omics techniques can spread into other BQEs, such as diatoms or benthic macroinvertebrates. Until today, in the case of proteomics, these techniques have only been carried out at the species level and not in communities [33,91]. Traditional studies based on the morphological determination of individuals have been disrupted by the arrival of HTS technologies, which have marked an evolutionary line of standard applications, ultimately converging on omics techniques. This evolution line encompasses the use of eDNA in conservation and biomonitoring. From single-marker analyses of species or communities to metagenomic surveys of entire ecosystems, it provides a predictive feature in spatial and temporal endangered freshwater biodiversity patterns [151]. Through omics techniques, genetic biomarkers, and diversity, we can gain insight into aquatic species’ adaptability, resistance, and sensitivity characteristics. This improves our mechanistic understanding of how stressors affect individuals and may lead to the development of predictive models to estimate their susceptibility to stressors [57]. Such as the movement of individuals between polluted and clean sites is a complex dynamic that involves adaptability, resistance, and sensitivity patterns of organisms and can be explored by ecotoxicological studies. Nonetheless, the most important understanding is the process developed in freshwater ecosystems at the various levels of biological organization based on the responses of living organisms to contaminants of emerging concern (CECs) [35]. For this, it is imperative to propose and structure new methodologies to merge molecular and ecological research. Less attention should be paid to designing studies around technical limitations [152].

### 3.2. Omics in a Scenario of More Severe Impacts

During the last decade, unusual issues that affect aquatic ecosystems and others that are already well acknowledged have increased in acuteness and spatiotemporal enlargement in their ecological processes. For the first term, for instance, natural and artificial nanocolloids that a xenologist conducts in freshwater environments (toxic impact on endogenous molecular mechanisms), and with omics techniques, they can understand these aforementioned mechanisms more deeply [153]. Secondly, stressors such as drought can shift the dissolved organic carbon (DOC) and other dissolved compounds in water. This provokes humic substances that impair macrophyte growth, which presents another opportunity for omics to analyze stress tolerance [154]. Despite the worrisome stressor for biomonitoring goals being the decline in water quality, water quantity destabilization also puts aquatic biota at risk. Functional traits such as sensibilities to current velocity or dissolved oxygen changes [155] are affected. Water deficits (caused by drought or water abstraction) lead to negative impacts on breeding macrophytes. In this sense, technological tools such as genomics can directly mitigate these stress effects by emphasizing the maintenance of genes associated with fitness [156]. This is important because the maintenance of macrophytes, water quality, and different sediment sizes in streams with frequent drying is a valid remediation strategy for biota refugees [157]. Being a macrophyte with diverse architecture serves as a refuge and heterogeneity of habitats for multiple zooplankton and macroinvertebrate organisms [158,159] and for the preservation of epiphytic biofilm [160]. Returning to recently evidenced stressors, nanoparticles provided by engineered nanoparticles, with similar characteristics to nanocolloids, have already been detected in freshwater ecosystems, generating potential environmental risk in, for example, plant litter decomposition processes [161]. Unraveling the diversity of microbial decomposers in streams is crucial to fully understand the dynamics of leaf litter and recalcitrant compound degradation. This is achievable through the omics approach, which allows us to discover functional traits in microbial diversity [162].

Speaking about fitness, the genotype and phenotype of certain organisms have been modified by genetic engineering. In the case of transgenic fish, metabolomic analysis of the hormonal growth of *Coho salmon* is also being conducted [163,164]. Furthermore, the sperm motility of endangered and threatened fish species is a possible way to investigate, through proteomics analysis, the fertilizing capacity and, therefore, the fitness variability [165]. Generally, there are promising advantages in using transgenics in fish through omics techniques. For example, modifications of sperm motility or hormonal metabolisms have been registered, as well as manipulation of pigmentation and fish body coloration [166]. From the standpoint of freshwater biomonitoring, transgenic organisms (*Daphnia* is the most significant model because they are more sensitive than fish) for heavy metal monitoring are straightforward. Still, their use is limited concerning detection studies [167]. Fitness impediments due to endocrine-disrupting chemicals (EDCs) have been analyzed in monitoring frameworks through biomarker candidates obtained by transcriptomic analysis, including certain orphan genes of *Daphnia pulex* or by proteomics in gammarids [168]. One of the most worrisome and chronic impacts on the fitness of organisms is exposure to metals [169], which is directly linked to freshwater biomonitoring. The aforementioned issue has been explored by [170], whose proposal aims to evaluate the current knowledge of microbes (bacteria, archaea, and protists) as bioindicators in routine assessments through environmental genomics, which would improve our understanding of ecosystem functions. Fitness, expressed by growth and survival, is shaped by maintaining ion and water balance, whose changes are controlled by total dissolved solids (TDS), especially in benthic macroinvertebrates [171]. For this particular matter, microalgae and cyanobacteria from wastewaters suffer from several limitations, with TDS being one of them [172]. Currently, in freshwater environments affected by disproportionate urbanization and industrialization, TDS and other physical and chemical properties of water are altered. This scenario negatively impacts the fitness of organisms. Demonstrating that it has promising practical feasibility and is a sustainable solution, phytoremediation (using aquatic plants) [173,174] and phycoremediation (using diatoms) [175], improved with omics techniques, have been researched and recommended.

Omics and bioengineering have been related in numerous cases, mainly in genome editing [176]. Although in freshwater biomonitoring, that link has been scarcely applied. In the case of Clustered Regularly Interspaced Short Palindromic Repeats/Cas proteins (CRISPR/Cas), a method integrated into genomics is used to provide new platforms for detecting cryptic species in harmful algal and cyanobacterial blooms [177]. CRISPR/Cas, in conjunction with eDNA-based studies (and metagenomics in some cases), is a promising tool for biodiversity assessments. Primary explorations have shown interesting projections [178]. CRISPR/Cas would be an efficient technology utilized in aquatic species related to public health. However, in the case of insects (mostly benthic invertebrates), there is a lack of systematic information about genome editing and its related goals [179]. Biomonitoring related to issues with public health is a research area that must have more suitable mechanisms within environmental policies, which is possible to achieve through cyberbiosecurity frameworks allowing for new technology requests (i.e., genomics practices combined with machine learning) [180] integrated into CRISPR/Cas when applicable. A comprehensive probable mechanism integrates omics techniques with bioengineering, linked by cyberbiosecurity, with the general purpose of recognizing risk assessment for public health (pre-, during, and post-disturbance stages). This is based on biomonitoring programs focused on determining biodiversity functions. For example, pre-disturbance events such as the emergence of recalcitrant toxic materials in water (such as per- and polyfluoroalkyl substances), are detectable through the maintenance of detection procedures using ecotoxicological laboratory methodologies. For more details on these specific compounds, see the review conducted by [181]. In post-disturbance (or during) stages, the incorporation of model species of organisms that have resistance traits (such as diapause) in rivers with drying episodes is preponderant [182]. In this context, these model species help to recognize recolonization and biotic continuity patterns and to monitor the regulation of chemical conditions and biodiversity of aquatic ecosystems affected by extreme drying by studying both aquatic and terrestrial species together.

One of the most direct advantages of using omics techniques in freshwater biomonitoring practices is the extension of molecular-based databases. It is necessary to conduct more sequencing studies to make genomes available to understand organisms’ responses to increasingly severe impacts. The assimilation of molecular-based datasets, conjugated with functional traits, is a potential general overview of the ecological features, which comprises traditional and modern (omics) techniques of systematization [183]. Namely, the preferences of stream invertebrate families for particular physical and chemical conditions, outlining behavioral characteristics such as hydroaerophily, rheophily, and thermophily, can be interrelated to omics information. In the case of aquatic microbiomes, which are relevant due to the conserved essential genes of bacteria, the molecular data are scattered across different studies, which does not allow for optimal curation and standardization [184]. In fact, there is already a database focused on gene expression studies in *Daphnia* that are strictly specific to stressor dynamics. It evaluates gene experiments due to environmental perturbations [185]. Moreover, overall, for cyanobacteria, there is a comprehensive and integrated web database related to omics data, such as genomic data (928 strains), independent transcriptomic information (56 datasets), primary (three datasets), and proteomic information (15 datasets) [186]. For diversity surveys, freshwater microorganisms have more significant potential for database formation due to the high amount of knowledge achieved regarding biogeochemical cycles, regulation of ecosystem services, etc. [187]. Additionally, for fish, there is a multi-level omics database that includes, among others, 233 fish genomes, 201 fish transcriptomes, and 5841 fish mitochondrial genomes [188]. What is also remarkable is the utilization and annotation of expressed sequence tags (ESTs) as a comparative genomics platform for threatened freshwater species, as is the case of the essential gene database of the bivalve *Pisidium coreanum* [189]. Biomonitoring must be conducted in a long-term and systematic manner, and the omics technologies must progress to biodiversity database augmentation, developed for taxonomic identification, together with artificial intelligence for collecting climatic, geological, hydrological, and other environmental data [190].

Currently, the challenges that threaten freshwater ecosystems and ecosystem/human health are unexplored, in part due to the lack of a holistic or multi-dimensional depiction of the biotic processes involved. In this matter, single-omics and multi-omics have been compared to better understand the metabolic processes of aquatic organisms under various environmental impacts. This includes factors such as sensitivity, health status, response, tolerance, and adaptation. [191]. The combination of single-omics and multi-omics opens the possibility of an immense spectrum of conceptual works, particularly in microbial ecology. However, microorganism interaction processes, especially periphyton, have been neglected [192], despite their potential importance to biomonitoring methods. With the advancement of omics techniques in the microbial community, the concepts derived from these studies (which are more numerous than other BQEs) could help to strengthen methods related to fishes or macroinvertebrates. This would lead to a more comprehensive and permeable biomonitoring approach using omics-type techniques. These tools working together could be a favorable approach towards the modernization of freshwater biomonitoring since most of the biological levels and the interaction settings between them can be tackled. Isotopomics could be a direct extension of omics techniques into trace signals on certain processes associated with the optimal status of ecological functions, such as photosynthesis rate [193] or food web structure [194]. Nonetheless, isotopomics from traditional isotopic analysis executed in ecology differ greatly depending on the internal or external evaluation of metabolic backgrounds. Omics techniques linked to multiple ecological concerns (so-called “eco-omics”) can shed light on the understanding of mechanistic relationships regarding gene-environment interactions through the generation of new hypotheses [195]. Therefore, the expected progress or strategy for more adaptable freshwater biomonitoring programs, with an important component of omics techniques, is to build upon the knowledge (concepts and methods) already acquired from studies on microorganisms and extrapolate it to other BQEs. These concepts and methods are related to standardization and protocol assessments to improve the steps of acquisition and understanding while maintaining the same comparability rate as fish or macroinvertebrates, for example.

### 3.3. Omics Data and Informatics Tools

Omics data and consequent informatics tools utilization are likely the most challenging aspects in adapting omics technologies to biomonitoring schemes. As mentioned previously, with the advancement of technologies based on NGS and microarrays, the use of chromatography, and mass spectrometry, among others, has made it possible to analyze and profile various biomolecules that provide a complete picture of cellular biology and function [196]. This has allowed us to understand the actual health status of ecosystems, thanks to the study of biomarkers frequently used in the analysis and biomonitoring of freshwaters [197]. Traditionally, to search for connections with biological processes, each data type has been examined separately. Using these techniques, we have assembled some parts of the genetic architecture of complex traits and fundamental biological pathways. Nonetheless, a large portion of the genetic basis for complex characteristics and biological networks is still unknown, which could be, in part, due to the emphasis on restrictive single-data-type study designs [198]. Many efforts have been made to analyze multi-omics data that provide reliable and valid information in freshwater ecology studies. In the study by [133], different omics techniques, including bacterial metagenomics and metabolomics, were integrated with physicochemical data (trace metals and water quality) using a data reduction methodology based on the intrinsic approach. Another study [199] used SIMCA-P12.0 for the analysis of multivariate data from nuclear magnetic resonance (NMR)-based metabolomics and metallomics, which are based on elemental distributions and interactions, to study microalgae, specifically *Chlorella* sp., in multimetal systems. The results showed that the two techniques complemented each other clearly, as the metabolomics data revealed biochemical changes generated by exposure to metals, while the metallomics approach revealed the flow of nutrients and heavy metals over time. A study by [200] used pairwise association and integration analysis of three omics data (transcriptomics, proteomics, and metabolomics).

The use of applications and websites that allow for the analysis of unique omics data has been widely described, such as the NCBI database, LIPIDSMAPS, Ingenuity Pathway Analysis, Cytoscape, etc. [201,202]. However, researchers have stated that analyzing single-type data is insufficient to understand biological systems, as many levels are used to regulate a system [203]. Data integration is the process by which different omics data are combined as predictor variables to enable more complete and complex modeling. This will allow us to explore new scientific questions about aspects of mutation, expression, and regulation in biology. However, this is not an easy task, as the diversity of size, missing patterns, noise across different data types, and the correspondence and correlation between measurements from different technologies are still a substantial challenge [198]. 

For this reason, categorized data integration through a meta-dimensional analysis has three approaches: concatenation-based integration, which combines multiple data matrices for each sample into a large input matrix before constructing the model; transformation-based integration, which combines various datasets after transforming each data type into an intermediate form, such as graphs or kernel matrices; finally, these are merged into multiple graphs to generate the model and produce model-based integration, based on generating multiple models using the different data types as training sets and then generating a final model from the multiple models created during the training phase. Machine learning analysis has become increasingly popular in biological sciences, including processing complex omics data and their integration. This method aims to acquire and learn from historical or current data to make predictions or select measures for future data. Machine learning algorithms have different attributes, so it is crucial to choose an appropriate algorithm for analysis, which is explained in detail in the work by [204]. In this sense, it is of utmost importance to establish an integrated platform that ensures the correct analysis of data obtained from these omics techniques. This allows for easy study of interactions between different omics. However, a multidisciplinary platform for the multi-omics analysis of aquatic ecosystems has not yet been achieved. Currently, there are analysis packages in R that provide semi-supervised methodologies such as mixOmics. This software allows for interaction and association between datasets, providing a unique data matrix. Additionally, a package called MiBiOmics has been implemented, allowing for the parallel study of up to three omics datasets and deep exploration and integration of each dataset, which is not provided by mixOmics [106]. As a synoptical view, Figure 1 illustrates omics data and informatics tools for each omics and multi-omics approach and system of data processing.

## 4. Conclusions

The utilization of technically advanced and multifaceted omics approaches for biodiversity assessments has the potential to significantly enhance freshwater biomonitoring. This is particularly important for improving the management of public health, ecosystem function, and sustainability while reducing the socioeconomic risks currently jeopardized by climate change and anthropic pressures. The extensive range of concepts and methodologies achieved through the utilization of metabarcoding procedures, microbial biology, and ecotoxicology serve as the foundation for the innovative achievements of omics. These accomplishments have been comprehensively described and elucidated in the present review. Utilizing bioindication methods, non-destructive techniques, and big data management, among other processes, has resulted in significant improvements in spatial and temporal efficiency. This applies to the consideration of BQEs separately or in combination with two or more of these. A comprehensive roadmap outlining the sequential steps involved in field-based and experimental protocols is imperative. This roadmap should consider the sampling, preservation, and extraction stages of biomolecules in field-based protocols, as well as the variability and innovation of experimental protocols. Integrating traditional and molecular biomonitoring techniques feasibly and systematically is recommended to address gaps in ecological indicators. This approach has been demonstrated in numerous studies and can contribute to a more comprehensive understanding of ecological systems. Integrating omics and traditional techniques can enhance the assessment of ecosystem health by combining established methodologies with more advanced and refined approaches. This approach allows for a comprehensive evaluation of the spectrum of internal bioactivity interactions that occur across various biological and ecological levels, which are often hierarchical in nature. It is imperative to establish additional model species for genomic, proteomic, and metabolomic analyses across all groups of BQEs. Standardization of database extension is a crucial element in addressing the primary disturbance across various ecoregions such as Mediterranean, temperate, etc., hydrological systems including streams, rivers, estuaries, lakes, etc., and countries categorized as developed or developing. In situations involving intricate and novel pollutants, the analysis of omics through bioengineering systems often relies on assessing fitness detriment. Such analysis must be conducted under the supervision of scientific constructs supported by ethical considerations.

## Figures and Tables

**Figure 1 biology-12-00923-f001:**
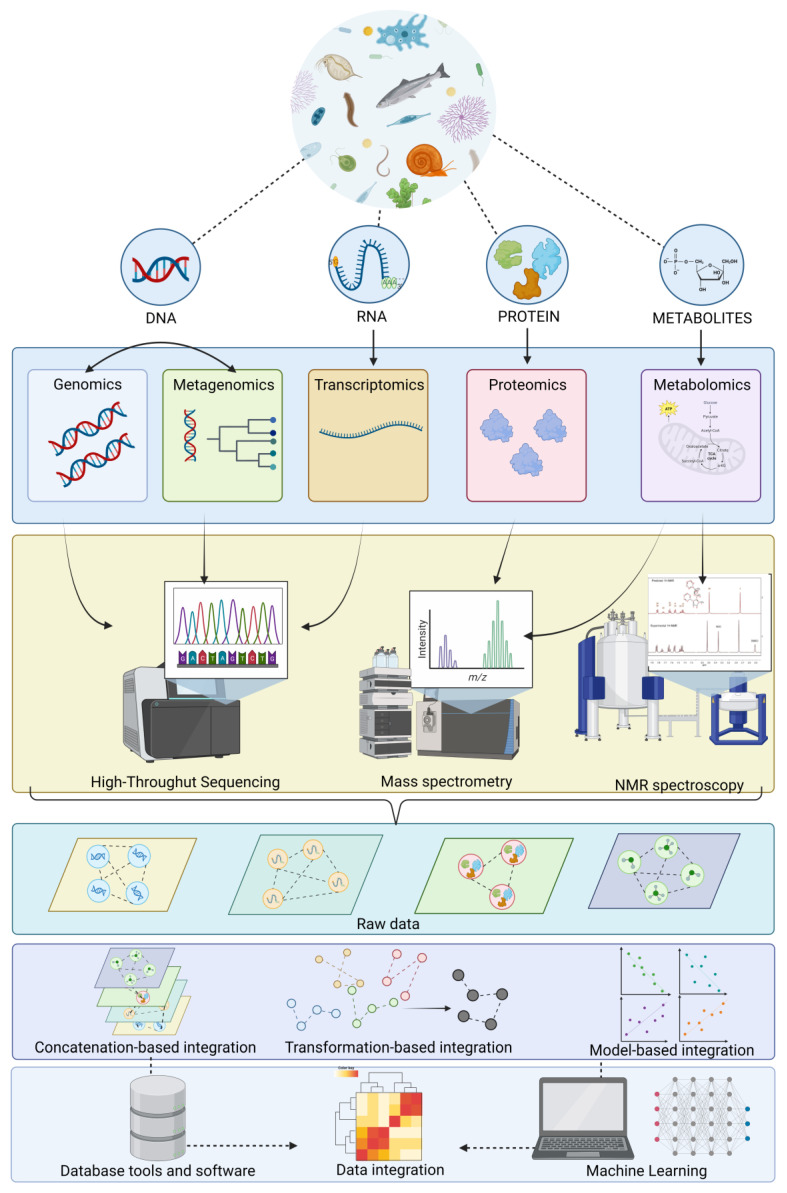
Schematic diagram of the workflow of the integrative multi-omics approach used in aquatic model animals to ensure reliable environmental risk assessment in freshwater ecosystems.

**Table 1 biology-12-00923-t001:** Advantage, disadvantage, and key finding of the individual omics technologies described in this review.

Technique	BQEs or Other as Example	Key Finding	Advantages	Disadvantages	References
Genomics	Diatoms	Seasonal carbon dynamics studies using photosynthesis rate.	Comparative genomics in ecotoxicology has identified new molecular biomarkers in species that have not yet been defined.	Lack of data on freshwater organisms in digital repositories, limiting progress in the field.	[76,77,78,79]
	Fish	Gene expression with many chemical/pharmaceutical stressors.			
Metagenomics	Macroinvertebrates	Study of individual biomass in communities.	Provide an overview of metabolic potential and reconstruct whole genome sequencing to understand taxa with small genomes.	Pathway genes or markers have a low frequency of recovery. Taxonomic assignments based on markers are rare. Large number of unknown genes.	[80,81,82]
	Diatoms	Estimation of biogeochemical processes in estuaries.			
Transcriptomics	*Dreissena* *polymorpha*	Ecotoxicological bioassessment of chemicals and insecticides.	Transcriptomic study is broad, and it is possible to study changes in environmental conditions, species adaptations, biomarkers, and alterations in metabolic pathways.	Transcriptomics can be associated with certain protein expression analyses that originate from oxygen related molecular responses. Lack of knowledge in non-coding RNAs.	[44,75,83,84,85,86,87,88]
	Catfish	Adaptation in high altitudes environments and their implications.			
	Aquatic insects	Life cycle adaptations and adaptative evolution in aquatic insects.			
	*Physa acuta* and *Chlamydomonas* (Green algae)	Search of new biomarkers by expressed sequence typing			
Proteomics	*Chlorella algae*	Variation in trophic chain due to exposure to metals.	More accurate in recognizing the effect of responses due to environmental conditions and selective pressures.	Requires the construction of more suitable databases for protein identification and pathway analysis in non-model species.	[33,89,90,91,92,93,94]
	Aquatic insect (Chironomidae)	Impact of natural insecticides on increased globin protein production.			
	*Daphnia*	Stress caused by temperature changes and exposure to microplastics that affect protein production.			
Metabolomics	*Daphnia*	Metabolite determination versus sublethal contaminant exposure.	It is capable to estimate biochemical metabolic changes by low molecular weight metabolites that represent the most functional measure of an organism’s physiology and response to toxic stress.	It is a technique rarely used in biomonitoring practices.	[46,95,96,97,98,99]
	*Scenedesmus obliquus*	Stress response in fullerenes conditions.			
	*Dreissena* *polymorpha*	Micropollutants effects (carbamazepina and methylmercury)			
	*Chloromonas* *augustae*	Interactive effects of stressors (Cu and heating)			
	*Stephanodiscus hantzchii*	Effect of temperature on internal metabolism.			
	Trichoptera larvae	Chemical contamination effect.			
Multi-omics	Algae	Metabolomic-transcriptomic, to measure the effect of the macrolide antibiotic clarithromycin on the reactions.	Can provide more information about mechanisms of action by which contaminants achieved adverse outcomes at higher levels of biological organization.	It is necessary to have an integrated platform that guarantees the correct analysis of the data obtained from the different omics.	[100,101,102,103,104,105,106]
	Microbial communities	Genomics-metagenomics, resistome exploration of the community			
	*Elodea nuttallii*	Transcriptomic-proteomic-metabolomic, asses the negative impact of methyl-Hg and inorganic Hg in food chains			

## Data Availability

Not applicable.

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
