# Peer review of "Current Status of Omics in Biological Quality Elements for Freshwater Biomonitoring"

_biology, 2023, doi:10.3390/biology12070923_

Round 1
Reviewer 1 Report
I think this is a good review paper that will help people understand the application of omics techniques to freshwater biomonitoring, including biodiversity and ecotoxicology.
I think that it is could be accepted after minor revison.
Line39: [1] are noticeably clear with respect to the modern conceptualization of freshwater bio- 39
Monitoring, [1]?? I don’t undestand this. Can you explain more clearly?
none
Author Response
Line39: “[1] are noticeably clear with respect to the modern conceptualization of freshwater biomonitoring?? I don’t understand this. Can you explain more clearly?”
R: The reference was misspelled, but it has already been fixed and explained (lines 52 and 53).
Reviewer 2 Report
Machuca-Sepulveda et al. review how omics in the environmental monitoring of freshwater ecosystems can be helpful by combining them with traditional methods. They review the different omics that can be used (genomics, metabolomics, etc.) and the interest in incorporating them into the current biomonitoring studies. Finally, the authors explain different situations and cases of interest where the combination could be relevant and the studies performed concerning this approach.
The review uses the bibliography properly and extensively revises the relevant issues on the topic. However, some reference is missed to the fact that the massive analyses are dependent on the database, which have noise, especially in the nucleotide sequences, increasing the difficulties in the analysis. In addition, the high number of unknown genes (especially in macroinvertebrates) and the lack of knowledge in non-coding RNAs in those animals make the approach more descriptive than explicative, requiring an effort to establish a validated database to perform the biomonitoring with a standard methodology.
Minor: The introduction seems to have a lost sentence at the start.
Author Response
“However, some reference is missed to the fact that the massive analyses are dependent on the database, which have noise, especially in the nucleotide sequences, increasing the difficulties in the analysis. In addition, the high number of unknown genes (especially in macroinvertebrates) and the lack of knowledge in non-coding RNAs in those animals make the approach more descriptive than explicative, requiring an effort to establish a validated database to perform the biomonitoring with a standard methodology.”
R: That information has already been added in Table 1.
“The introduction seems to have a lost sentence at the start”.
R: The sentence was misspelled, but it has already been fixed and explained (lines 52 and 53).
Reviewer 3 Report
The draft is very well structured and written. I congratulate the writers for the vast revision made and for the understanding that omic technology has and will have more and more comprehensive in the themes covered.
The content covered is so vast that I admit that I do not have the competence to judge the entirety of the darft, I would just suggest that in each topic addressed, you cite more examples of species (in a broad and non-local context, but of species, for example, that have the genome complete, or that are used for transcriptomas, among others)...
For example in the neotropical scenario we have recently the genome of the tambaqui (Colossoma macropomum), and there may be several other examples in several different realities.
Focused on my area of expertise, I would like to suggest stronger references to lipid metabolismo
Castro, L. Filipe C., Douglas R. Tocher, and Oscar Monroig. "Long-chain polyunsaturated fatty acid biosynthesis in chordates: Insights into the evolution of Fads and Elovl gene repertoire." Progress in lipid research 62 (2016): 25-40.
Xie, Dizhi, et al. "Regulation of long-chain polyunsaturated fatty acid biosynthesis in teleost fish." Progress in Lipid Research 82 (2021): 101095.
Monroig, Óscar, et al. "Desaturases and elongases involved in long-chain polyunsaturated fatty acid biosynthesis in aquatic animals: From genes to functions." Progress in lipid research (2022): 101157.
such situations can be added to the line 195
“At once, whole genome sequencing, transcriptomics, proteomics, and metabolomics have revealed a detailed mechanism of lipid catabolism and anabolism, which are related to multiple various structural and growth types [62]”
Author Response
“…you cite more examples of species (in a broad and non-local context, but of species, for example, that have the genome complete, or that are used for transcriptomes, among others)...”
R: Several species and assemblages have already been added in the manuscript, specifically in Table 1. Nonetheless, regarding genomes, species have been reported and referenced (lines 294 to 296).
“Focused on my area of expertise, I would like to suggest stronger references to lipid metabolism:
Castro, L. Filipe C., Douglas R. Tocher, and Oscar Monroig. "Long-chain polyunsaturated fatty acid biosynthesis in chordates: Insights into the evolution of Fads and Elovl gene repertoire." Progress in lipid research 62 (2016): 25-40.
Xie, Dizhi, et al. "Regulation of long-chain polyunsaturated fatty acid biosynthesis in teleost fish." Progress in Lipid Research 82 (2021): 101095.
Monroig, Óscar, et al. "Desaturases and elongases involved in long-chain polyunsaturated fatty acid biosynthesis in aquatic animals: From genes to functions." Progress in lipid research (2022): 101157.
such situations can be added to the line 195.”
R: Such references have already been incorporated (lines 209 to 214).